# Sinomenine Inhibits Migration and Invasion of Human Lung Cancer Cell through Downregulating Expression of miR-21 and MMPs

**DOI:** 10.3390/ijms21093080

**Published:** 2020-04-27

**Authors:** Kun-Hung Shen, Jui-Hsiang Hung, Yi-Ching Liao, Shu-Ting Tsai, Ming-Jiuan Wu, Pin-Shern Chen

**Affiliations:** 1Division of Urology, Department of Surgery, Chi Mei Medical Center, Tainan 710, Taiwan; robert.shen@msa.hinet.net; 2Department of Urology, Taipei Medical University, Taipei 110, Taiwan; 3Department of Biotechnology, Chia Nan University of Pharmacy & Science, Tainan 71710, Taiwan; hung86@mail.cnu.edu.tw (J.-H.H.); clairexiao75@gmail.com (Y.-C.L.); shuting831114@gmail.com (S.-T.T.); imwu@gm.cnu.edu.tw (M.-J.W.)

**Keywords:** sinomenine, lung cancer, cell invasion, matrix metalloproteinase, microRNA-21

## Abstract

Sinomenine is an alkaloid derived from *Sinomenium acutum*. Recent studies have found that sinomenine can inhibit various cancers by inhibiting the proliferation, migration and invasion of tumors and inducing apoptosis. This study aims to investigate the effect and mechanism of sinomenine on inhibiting the migration and invasion of human lung adenocarcinoma cells in vitro. The results demonstrate that viabilities of A549 and H1299 cells were inhibited by sinomenine in a dose-dependent manner. When treated with sub-toxic doses of sinomenine, cell migration and invasion are markedly suppressed. Sinomenine decreases the mRNA level of matrix metalloproteinase-2 (MMP-2), MMP-9, and the extracellular inducer of matrix metalloproteinase (EMMPRIN/CD147), but elevates the expression of reversion-inducing cysteine-rich proteins with kazal motifs (RECK) and the tissue inhibitor of metalloproteinase-1 (TIMP-1) and TIMP-2. In addition, sinomenine significantly increases the expression of the epithelial marker E-cadherin but concomitantly decreases the expression of the mesenchymal marker vimentin, suggesting that it suppresses epithelial–mesenchymal transition (EMT). Moreover, sinomenine downregulates oncogenic microRNA-21 (miR-21), which has been known to target RECK. The downregulation of miR-21 decreases cell invasion, while the upregulation of miR-21 increases cell invasion. Furthermore, the downregulation of miR-21 stimulates the expression of RECK, TIMP-1/-2, and E-cadherin, but reduces the expression of MMP-2/-9, EMMPRIN/CD147, and vimentin. Taken together, the results reveal that the inhibition of A549 cell invasion by sinomenine may, at least in part, be through the downregulating expression of MMPs and miR-21. These findings demonstrate an attractive therapeutic potential for sinomenine in lung cancer anti-metastatic therapy.

## 1. Introduction

Lung cancer is one of the most common types of malignant cancers in many countries. Non-small cell lung cancer accounts for approximately 85% of lung cancer with a generally poor prognosis and a high potential for metastasis [1]. For most patients with advanced and metastatic lung cancer, surgical resection may be not suitable, thus rendering cytotoxic treatments including chemotherapy and radiotherapy as major therapy strategies. Advanced lung cancer cells are highly invasive and result in high mortality rates. In order to improve the current therapeutic efficacy, developing more effective agents that target cancer cell invasion and metastasis is necessary.

Cancer metastasis is a complex process involving multiple steps, including the detachment of cells from the primary tumor, degradation of the extracellular matrix (ECM), invasion of cells to the surrounding tissue, entry into blood vessels or lymphatic vessels, and finally extravasation in new tissue [2,3]. Matrix metalloproteinases (MMPs) are important enzymes that contribute to tumor cell migration, invasion, and metastasis [4]. Of these MMPs, MMP-2 and MMP-9 are the major proteases that participate in the process of metastasis [5,6]. Activation of these enzymes is associated with increased tumor metastasis, which suggests a crucial function for these proteases in metastasis [7].

Degradation of the ECM is crucial for tumor invasion and metastasis. Several proteins are responsible for the ECM degradation and cell invasion, such as the extracellular inducer of matrix metalloproteinase (EMMPRIN/CD147), reversion-inducing, cysteine-rich proteins with kazal motifs (RECK) and tissue inhibitor of metalloproteinases (TIMPs). EMMPRIN/CD147 is highly expressed in various cancer cells and functions as an activator of proteinases including MMPs in tumor and neighboring stromal cells, and contributes to the regulation of the tumor microenvironment [8,9]. RECK is a membrane-anchored MMP inhibitor that negatively regulates activities of MMP-2 and MMP-9, and consequently reduces cancer invasion and metastasis [10]. A low expression of RECK is associated with a high invasiveness and poor prognosis [11]. TIMPs are endogenous tissue inhibitors that suppress activities of most MMPs. The balance between MMP and TIMP levels is a central determinant of the net proteolytic activity [12]. Tumor dissemination is also facilitated by the epithelial–mesenchymal transition (EMT) of tumor cells, through which cells exhibit greater motility and invasiveness [13,14]. The process of EMT is associated with the downregulation of epithelial markers such as E-cadherin, and the upregulation of mesenchymal markers such as vimentin [15].

MicroRNAs (miRNAs) represent a class of small non-coding RNAs that suppress the mRNA translation or stability by interacting with the target genes. miRNAs regulate multiple important biological processes including the cellular proliferation, differentiation, and pathogenesis of cancers [16,17]. Some miRNAs have been shown to serve as oncogenes or tumor suppressors [18,19]. An oncogenic miRNA, miR-21, is overexpressed in various tumors including lung, prostate, breast, pancreas, and colon [20,21]. miR-21 stimulates cancer cell proliferation, inhibits apoptosis, and increases invasion and metastasis by targeting multiple molecules such as programmed cell death (Pdcd4) [22], the phosphatase and tensin homolog gene (PTEN) [23], and RECK [24].

Sinomenine is an alkaloid extracted from *Sinomenium acutum* [25]. Previous studies have shown that sinomenine exhibits anti-inflammatory and anti-rheumatic effects [26,27]. In addition, sinomenine is effective in inhibiting proliferation and enhancing apoptosis of human cancer cell lines such as lung, gastric, liver, and breast cancers and glioblastoma [28,29,30,31,32]. Sinomenine also suppresses the migration and invasion of various human cancer cells [33,34,35,36]. Recent studies have demonstrated that sinomenine exerts anti-cancer activity by regulating various miRNAs. Sinomenine inhibits breast cancer cell migration and invasion by the modulation of miR-324-5p and miR-29 [33,35]. Sinomenine reveals anti-tumor activity by the induction of miR-204 in gastric cancer cells [37]. Moreover, sinomenine suppresses cell proliferation, migration and invasion by the downregulation of miR-23a [36].

Although sinomenine exerts anti-tumor potential against various cancer cell lines, the effect of sinomenine on miR-21-involved cancer cell migration and invasion remains unclear. This work determines the inhibitory effect and the molecular mechanisms of sinomenine on metastasis in vitro. A human lung adenocarcinoma cell is used in these experiments because of its highly invasive and metastatic characteristics. This study provides evidence that sinomenine can suppress the invasion of lung adenocarcinoma cells, suggesting a novel strategy for lung cancer treatment.

## 2. Results

### 2.1. Cytotoxic Effect of Sinomenine on A549 Cells

The chemical structure of sinomenine is shown in Figure 1A. The cytotoxic effect of sinomenine on human lung cancer A549 and H1299 cells is demonstrated in Figure 1B,C. The results reveal that treatment with 0.2 mM of sinomenine for 24 h significantly decreases the viability of A549 and H1299 cells, while treatment at doses below 0.1 mM does not cause cytotoxicity.

### 2.2. Effects of Sinomenine on Migration and Invasion of A549 Cells

In the view of cytotoxicity at a higher concentration of sinomenine, the inhibitory effect of non-toxic doses of sinomenine on the migration and invasion of A549 cells was investigated. After incubation with various concentrations of sinomenine for 24 h, 0.2 mM of sinomenine significantly suppresses the migration of A549 cells (Figure 2A,B). The inhibitory effect of sinomenine on the migration of H1299 cells was also observed (Figure 2C,D). These results demonstrate that sinomenine significantly inhibits the migration of A549 and H1299 cells.

In order to determine the inhibitory effect of sinomenine on the invasion of A549 cells across the extracellular matrix, the cells that invaded through the Matrigel-coated polycarbonate filter in the Boyden chamber were determined. The results show that sinomenine suppresses the invasion of A549 cells across the Matrigel-coated filter in a dose-dependent manner. Treatment with sinomenine at doses of 0.1 and 0.2 mM inhibited 26.5% and 40.8% of cell invasion, respectively (Figure 3A,B). The inhibitory effect of sinomenine on the invasion of H1299 cells across the Matrigel-coated filter was also observed (Figure 3C,D). These results indicate that sinomenine markedly inhibits the invasion of A549 and H1299 cells.

### 2.3. Sinomenine Decreases Expression of MMP-2, MMP-9, EMMPRIN/CD147 and Vimentin But Induces Expression of RECK, TIMP-1, TIMP-2 and E-Cadherin in A549 Cells

In order to investigate the mechanism of sinomenine on suppressing migration and invasion, A549 cells were used for the following experiments. During cell invasion, a proteolytic degradation of the ECM is required. Therefore, the effect of sinomenine on the expression of genes involved in the ECM degradation was analyzed by quantitative real-time PCR. The primer sequences are listed in Table 1. Data show that sinomenine decreases the mRNA expression of MMP-2, -9, and EMMPRIN/CD147 (Figure 4A). In addition, sinomenine enhances the expression of TIMP-1, -2, and RECK, which negatively regulate the activity of MMPs (Figure 4B). These results suggest that expressions of these genes involved in the degradation of the ECM are affected by sinomenine. Moreover, sinomenine significantly increases the expression of E-cadherin but decreases the expression of vimentin on the mRNA level in A549 cells (Figure 4C), suggesting sinomenine may induce the reversal of EMT.

### 2.4. Sinomenine Downregulates Expression of miR-21

Previous reports demonstrated that miR-21 is overexpressed in lung cancer cells in which miR-21 targets RECK and subsequently enhances cell invasion [20,24]. Quantitative real-time PCR was performed to detect the effect of sinomenine on the expression of miR-21. The result shows that sinomenine downregulates the expression of miR21 in a dose-dependent manner (Figure 5A). The decrease in the miR-21 expression by sinomenine may be involved in the anti-invasive mechanisms of sinomenine. To further investigate the role of miR-21 in the invasion of A549 cells, miR-21 mimics or inhibitors were transfected into A549 cells. The results demonstrate that cell invasion is elevated by the miR-21 mimics but reduced by the miR-21 inhibitors (Figure 5B). These findings indicate that the inhibition of the miR-21 expression may suppress cell invasion.

### 2.5. Silencing miR-21 Decreases Expression of MMP-2, MMP-9, EMMPRIN/CD147, and Vimentin but Induces Expression of RECK, TIMP-1, TIMP-2, and E-Cadherin in A549 Cells

To further investigate the effect of miR-21 on the expression of genes involved in cell invasion, quantitative real-time PCR was performed to detect the expression of genes after A549 cells were transfected with the miR-21 mimics or inhibitors. The results show that the miR-21 mimics decrease while the miR-21 inhibitors increase the expression of RECK, which is known as a target of miR-21. Moreover, the miR-21 inhibitors suppress the mRNA expression of MMP-2, -9, and EMMPRIN/CD147, but elevate the expression of TIMP-1/-2 (Figure 6A,B). The miR-21 inhibitors also increase the mRNA expression of E-cadherin but decrease the expression of vimentin (Figure 6C). These results reveal that the inhibition of the miR-21 expression may contribute to the expression of genes involved in the proteolytic activation and EMT, subsequently suppressing cell invasion.

## 3. Discussion

Sinomenine, the major bioactive alkaloid extracted from *Sinomenium acutum*, has been shown to possess anti-cancer activity against various cancer cell lines [29,30,31,32]. Our results demonstrate that sinomenine inhibits the viability of human lung cancer A549 and H1299 cells at the concentration of 0.2 mM. The results are consistent with a previous report that sinomenine induces cytotoxicity in A549 cells with an mM range [34]. Although the cytotoxic effect of sinomenine in the mM range has been reported in various cancer cell lines [30,33,34], the effect values in the μM range are reported. Sinomenine inhibits the viability of glioma U87 cells and clear-cell renal carcinoma ACHN cells in the μM range [38,39]. The sensitivity of the cytotoxic response of sinomenine is different in different cancer cell lines. The present findings demonstrate that sinomenine significantly suppresses the proliferation of human lung cancer A549 and H1299 cells. To elucidate the effect of sinomenine on cell motility, cell migration and invasion were determined by in vitro wound healing and Boyden chamber invasion assay, respectively. The present results reveal that sinomenine suppresses the migration and invasion of A549 and H1299 cells. In the Boyden chamber invasion assay, cells degrade the ECM components and perform transmigration across the ECM [40]. Pericellular proteolytic degradation of the ECM plays a critical role in tumor cell invasion, and is controlled by extracellular proteases, such as MMPs. Of these proteases, the expression of MMP-2 and MMP-9 are associated with lung cancer progression [41]. Previous reports also indicated that the downregulation of MMP-2 and MMP-9 inhibits the invasion and metastasis of lung cancer cells [41,42]. In view of this, whether the inhibitory effect of sinomenine on cell invasion is through the downregulating expression of MMPs was explored in A549 cells. The present results show that sinomenine suppresses the invasion and MMP-2/-9 expression in A549 cells, suggesting that sinomenine inhibits cell invasion through suppressing the MMP-2/-9 expression. The findings are consistent with previous reports showing that sinomenine inhibits invasion in various cancer cells through the downregulation of the MMP-2/-9 expression [36,43,44].

During tumor cell invasion, proteolytic degradation of the ECM is also regulated by various proteins. EMMPRIN/CD147 has been shown to activate MMPs, stimulate angiogenic factors in tumor and stromal cells, and modify the tumor microenvironment [8]. RECK negatively regulates the activity of MMPs and reduces cancer cell invasion, metastasis, and tumor angiogenesis [10]. The low expression of RECK in lung tumor tissues is associated with poor prognosis [45]. Recent reports implicated that the augmentation of the RECK expression can inhibit cancer cell invasion [46,47]. TIMPs act as the negative regulators of MMPs by which they are involved in the regulation of the invasion, metastasis, and angiogenesis of various cancers [48,49]. The upregulation of TIMP-1 contributes to inhibition of cancer cell invasion [50,51]. The results of this study suggest that the anti-invasive effect of sinomenine may attribute to the inhibition of the MMP-2/-9 and EMMPRIN/CD147 expressions, and the induction of the RECK and TIMP-1/-2 expressions.

The EMT of tumor cells plays an important role in cell migration and invasion. Stimulation of an EMT can cause cancer cells to invade the surrounding stroma, so a reversal of the EMT is thought to be an effective strategy against cancer metastasis [16]. This study observed the capability of sinomenine in reversing an EMT by downregulating the mesenchymal marker vimentin, and upregulating the epithelial marker E-cadherin. The results suggest that sinomenine induces the reversal of an EMT and may subsequently contribute to the inhibition of invasion in A549 cells.

Many miRNAs have been shown to regulate tumor initiation, invasion, metastasis, and angiogenesis [19]. Overexpression of oncomir miR-21 has been observed in many types of human cancers in which miR-21 promotes tumor initiation and invasion [20]. A recent study showed that miR-21 enhanced the invasion of lung cancer cells by targeting RECK, a negative regulator of several MMPs [24]. The present results demonstrate that sinomenine increases the RECK expression but decreases the miR-21 expression. In addition, the miR-21 mimics downregulate while the miR-21 inhibitors upregulate the expression of RECK. These findings imply that sinomenine stimulates the RECK expression through the downregulation of miR-21.

In order to elucidate whether the expression of MMP-2/-9, EMMPRIN/CD147, TIMP-1/-2, E-cadherin, and vimentin by sinomenine is regulated by the expression of miR-21, the miR-21 mimics and inhibitors were transfected to A549 cells and the gene expression was determined. The results demonstrate that the miR-21 inhibitors suppress the expression of MMP-2/-9, EMMPRIN/CD147, and vimentin, but elevate the expression of TIMP-1/-2 and E-cadherin. However, there is no evidence showing that miR-21 can directly target these genes. Therefore, miR-21 may regulate other pathways, which in turn affect the expression of the genes involved in cell invasion. Recent studies have demonstrated that miR-21 activates the PI3K/Akt signaling pathway by targeting PTEN, a negative regulator of the PI3K/Akt pathway [52,53]. Previous reports have indicated that the expression of the matrix metalloproteinases gene is mediated by the PI3K/Akt pathway [54,55]. In addition, miR-21 regulates the EMT in lung cancer cells through the PTEN/Akt signaling pathway [56]. Thus, miR-21 may affect the expression of MMP-2/-9, EMMPRIN/CD147, TIMP-1/-2, E-cadherin, and vimentin through mediating the PI3K/Akt/PTEN pathway.

NF-κB is crucial for the invasion and metastasis of some cancers [57,58]. Several naturally occurring anti-cancer agents inhibit migration and invasion in various cancer cell lines through the downregulation of NF-κB [59]. A previous report also demonstrated that sinomenine inhibits breast cancer cell invasion and migration by suppressing NF-κB [33]. In addition, NF-κB is essential for an EMT, which is considered to be closely associated with invasion and metastasis [13]. In the present study, results show that sinomenine regulates the expression of MMP-2/-9, EMMPRIN/CD147, TIMP-1/-2, E-cadherin, and vimentin. Sinomenine may affect the expression of those genes by regulating the activity of NF-κB. For a further understanding the role of NF-κB on the anti-cancer effect of sinomenine, the effect of sinomenine on the activity of NF-κB will be investigated in the future. The inhibitors and specific siRNA of NF-κB (p65) will be used to elucidate the effect of sinomenine. Furthermore, an investigation of the anti-metastatic effect of sinomenine will be carried out in an animal model in the future. The therapeutic activity and pharmacodynamic characteristics of sinomenine in vivo will also be performed.

## 4. Materials and Methods

### 4.1. Reagents and Cell Culture 

Sinomenine, dimethyl sulfoxide (DMSO), Tris-HCl, and EDTA were purchased from Sigma-Aldrich (St. Louis, MO, USA). A powdered Dulbecco’s modified Eagle’s medium (DMEM) was purchased from Gibco/BRL (Gaithersburg, MD, USA). Matrigel was purchased from BD Bioscience (Franklin, NJ, USA). The total RNA extraction kit and PCR kit were from Invitrogen (Carlsbad, CA, USA). The FastStart Universal Probe Master assay kit was purchased from Roche Applied Science. The human lung adenocarcinoma cell line A549 was obtained from BCRC (Food Industry Research and Development Institute, Hsinchu, Taiwan). The human non-small cell lung carcinoma cell line H1299 was a gift from Dr. Ying-Jan Wang (Department of Environmental and Occupational Health, National Cheng Kung University, Tainan, Taiwan). Cells were maintained in the DMEM supplemented with 10% fetal calf serum, 100 U/mL of penicillin, and 100 µg/mL streptomycin, and incubated in a 5% CO_2_-humidified incubator at 37 °C. For the sinomenine treatment, sinomenine was dissolved in DMSO and diluted with a culture medium (the final concentration of ethanol was less than 0.2%).

### 4.2. Cell Viability Assay 

The assay was performed as described previously [60]. Briefly, 3 × 10^3^ cells were seeded in a 96-well plate and treated with sinomenine in triplicate. After 24 h of incubation, the medium was replaced with a fresh medium containing 0.5 mg/mL MTT [3-(4,5-dimethylthiazol-2-yl)-2,5-diphenyltetrazolium bromide]. After 4 h, the supernatants were removed and the resulting MTT formazan was solubilized in DMSO and measured spectrophotometrically at 570 nm.

### 4.3. Wound Healing Migration Assay 

The assay was performed as described previously [61]. A549 cells were plated in a 12-well plate and grew to confluence. The monolayer culture was then scrape-wounded with a sterile micropipette tip to create a denuded zone (gap) of a constant width. After removing the cellular debris with PBS, cells were exposed to various concentrations of sinomenine for 24 h. A549 cells that migrated to the wounded region were observed by an Olympus CK-2 inverted microscope and photographed (100× magnification). The wound area was measured by the program Image J (http://rsb.info.nih.gov/ij/). The percentage of wound closure was estimated by the following equation: Wound closure % = [1 − (wound area at T_t_/wound area at T_0_) × 100%], where T_t_ is the time after wounding and T_0_ is the time immediately after wounding.

### 4.4. Boyden Chamber Invasion Assay

The Boyden chamber invasion assay was carried out as previously described [56]. Briefly, the polycarbonate filter (8 µm pore) was pre-coated with Matrigel. After being treated with sinomenine for 24 h, the cells (6 × 10^3^ cells/well) were added to the upper chamber in a serum-free medium. The complete medium (containing 10% FBS) was applied to the lower chamber as a chemoattractant. The chamber was incubated for 6 h at 37 °C. At the end of incubation, the cells in the upper surface of the membrane were carefully removed with a cotton swab and the cells that invaded the lower surface of the membrane were fixed with methanol and stained with a 5% Giemsa solution. The invaded cells on the lower surface of the membrane filter were scored from five random fields under microscopy (200× magnification).

### 4.5. Gene Expression Analysis

The total RNA was extracted using a TRIzol reagent (Invitrogen, Carlsbad, CA, USA) according to the manufacturer’s instructions. The total RNA (1 µg) from each sample was subject to reverse transcription with oligo (dT) primers by a PCR kit according to manufacturer’s instruction. The mRNA expressions of MMP-2, -9, EMMPRIN/CD147, TIMP-1,-2, and RECK were determined by quantitative real-time PCR which was conducted in the StepOne system (Applied Biosystem, Foster City, CA, USA). Briefly, each amplification mixture (50 μL) contained 10 ng cDNA and 25 μL SYBR Green PCR Master Mix. The PCR conditions were as follows: 95 °C for 2 min, 40 cycles at 95 °C for 15 s, and 60 °C for 45 s. For the detection of miR-21, a FastStart Universal Probe Master assay kit (Roche Applied Science, Penzberg, Germany) was used according to manufacturer’s instruction, and RNU6B was used as the internal control. The primer sequences are listed in Table 1. The PCR results were derived using the comparative C_T_ method. The chemically synthesized miR-21 mimics, miR-21 inhibitors, and the corresponding scramble negative controls were purchased from Quantum Biotechnology (Taichung, Taiwan). A549 cells were transfected with 50 nM miR-21 mimics, or 30 nM miR-21 inhibitors by the Lipofectamine 2000 (Invitrogen, Carlsbad, CA, USA).

### 4.6. Statistical Analysis

The data were expressed as mean ± standard deviation. The statistical significance was analyzed by a one-way ANOVA. If the significance was observed, the Dunnett’s post-hoc test was used to determine the difference between the treatment groups and the untreated group, with values of *p* < 0.05 considered statistically significant.

## 5. Conclusions

In conclusion, the present results reveal the anti-invasive effect of sinomenine in human lung cancer cells. The regulation in the expression of MMP-2/9, EMMPRIN/CD147, TIMP-1/-2, RECK, E-cadherin, and vimentin by sinomenine is possibly caused by the expression of miR-21. The effect of sinomenine on regulating the expression of these genes and miR-21 may result in the inhibition of invasion in lung cancer cells. The present study is the first report that indicates sinomenine downregulates the expression of miR-21, by which sinomenine suppresses migration and invasion in human lung carcinoma cells. The current findings disclose a new therapeutic potential of sinomenine in lung cancer therapy.

## Figures and Tables

**Figure 1 ijms-21-03080-f001:**
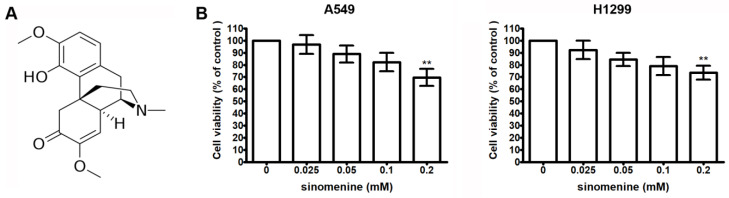
Sinomenine inhibits viability of human lung cancer cells. (**A**) Chemical structure of sinomenine. (**B**) Effect of sinomenine on viability of A549 and H1299 cells. Cells were treated with various concentrations of sinomenine for 24 h. Cell viability is presented as mean ± S.D. of four independent experiments. ** *p* < 0.01 compared with the untreated control.

**Figure 2 ijms-21-03080-f002:**
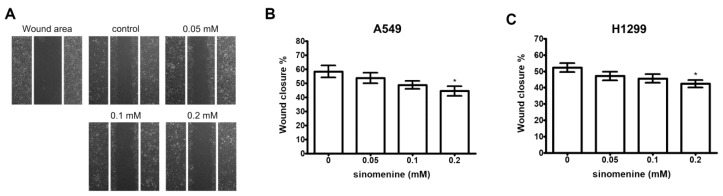
Effect of sinomenine on migration of A549 and H1299 cells. (**A**) A549 cell monolayers were scraped by a sterile micropipette tip and the cells were treated with various doses of sinomenine for 24 h. Cells that migrated to the wounded region were photographed (100× magnification). The wound area of the cultures of A549 cells (**B**) and H1299 cells (**C**) were quantified in four fields in each treatment, and data were calculated from three independent experiments. Data are presented as mean ± S.D. of three independent experiments. * *p* < 0.05 compared with the untreated control.

**Figure 3 ijms-21-03080-f003:**
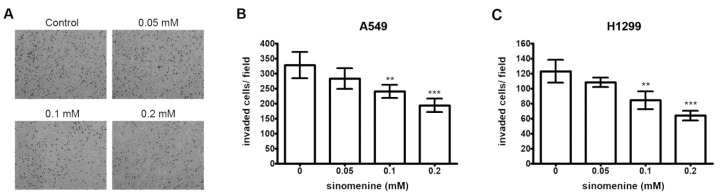
Effect of sinomenine on the invasion of A549 and H1299 cells. (**A**) A549 cells were treated with various concentrations of sinomenine for 24 h and cell invasion assay was performed. The invaded cells were photographed (200× magnification). The invaded A549 cells (**B**) and H1299 cells (**C**) were counted in five random fields in each treatment, and data were calculated from three independent experiments. Data are presented as mean ± S.D. of three independent experiments. ** *p* < 0.01, *** *p* < 0.001 compared with the untreated control.

**Figure 4 ijms-21-03080-f004:**
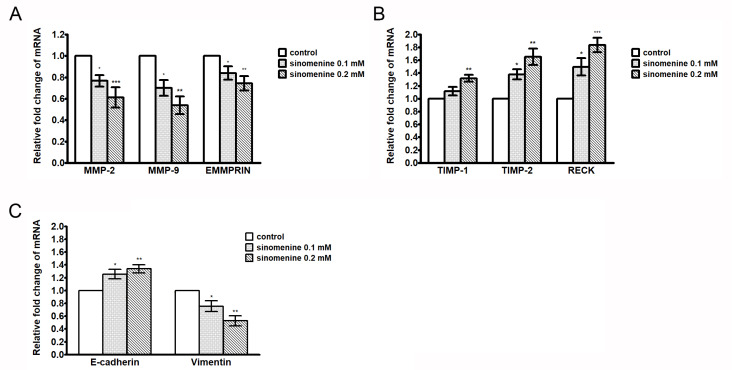
Effect of sinomenine on the expressions of MMP-2/-9, EMMPRIN/CD147, TIMP-1/-2, RECK, E-cadherin, and vimentin in A549 cells. Cells were treated with various concentrations of sinomenine for 24 h and the expressions of MMP-2, MMP-9, and EMMPRIN/CD147 mRNA (**A**), TIMP-1, TIMP-2, and RECK (**B**), and E-cadherin and vimentin (**C**) were analyzed by quantitative real-time PCR. Data are expressed as mean ± S.D. of three independent experiments. * *p* < 0.05, ** *p* < 0.01, *** *p* < 0.001 compared with the untreated control.

**Figure 5 ijms-21-03080-f005:**
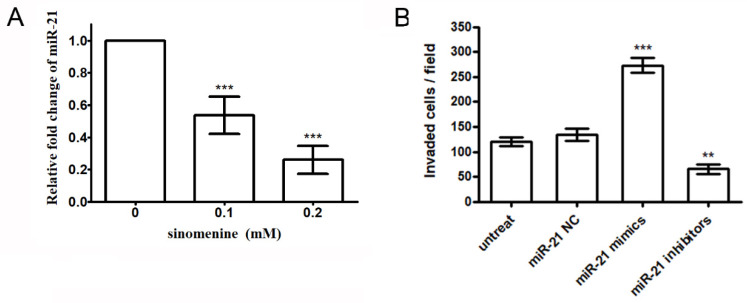
Effects of sinomenine on the expression of miR-21. (**A**) Cells were treated with various concentrations of sinomenine for 24 h and the expressions of miR-21 were analyzed by quantitative real-time PCR. (**B**) A549 cells were transfected with the miR-21 negative control (miR-21 NC), miR-21 mimics, or mir-21 inhibitors for 48 hrs. Cell invasion was performed by Boyden chamber invasion assay. Data are expressed as mean ± S.D. of three independent experiments. ** *p* < 0.01, *** *p* < 0.001 compared with the untreated control.

**Figure 6 ijms-21-03080-f006:**
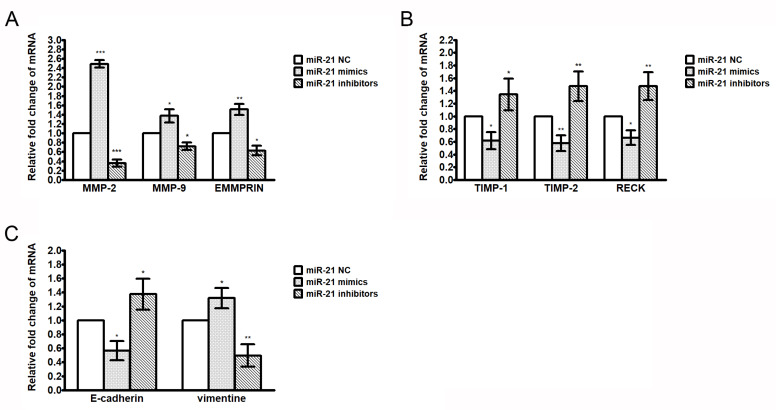
Effect of miR-21 on the expressions of MMP-2/-9, EMMPRIN/CD147, TIMP-1/-2, RECK, E-cadherin, and vimentin in A549 cells. A549 cells were transfected with the miR-21 negative control, miR-21 mimics, or mir-21 inhibitors for 48 hrs. Cells were harvested and the expressions of MMP-2, MMP-9, and EMMPRIN/CD147 mRNA (**A**), TIMP-1, TIMP-2, and RECK (**B**), and E-cadherin and vimentin (**C**) were analyzed by quantitative real-time PCR. Data are expressed as mean ± S.D. of three independent experiments. * *p* < 0.05, ** *p* < 0.01, *** *p* < 0.001 compared with the untreated control.

**Table 1 ijms-21-03080-t001:** Primer pairs used in quantitative real-time PCR.

Gene	Sequence (5′-3′)
MMP-2-F	CTTCCAAGTCTGGAGCGATGT
MMP-2-R	TACCGTCAAAGGGGTATCCAT
MMP-9-F	GGGACGCAGACATCGTCATC
MMP-9-R	TCGTCATCGTCGAAATGGGC
EMMPRIN-F	CTACACATTGAGAACCTGAACAT
EMMPRIN-R	TTCTCGTAGATGAAGATGATGGT
RECK-F	CCTGCATTGCTCGCTGTGTG
RECK-R	CCTGTGGTTTGGGTATGCACCTT
TIMP-1-F	CTTCTGCAATTCCGACCTCGT
TIMP-1-R	CCCTAAGGCTTGGAACCCTTT
TIMP-2-F	AAGCGGTCAGTGAGAAGGAAG
TIMP-2-R	CACACACTACCGAGGAGGG
β-actin-F	CATGTACGTTGCTATCCAGGC
β-actin-R	CTCCTTAATGTCACGCACGAT
E-cadherin-F	ACCAGAATAAAGACCAAGTGACCA
E-cadherin-R	AGCAAGAGCAGCAGAATCAGAAT
vimentin-F	AATGGCTCGTCACCTTCGTGAAT
vimentin-R	CAGATTATG TTCCCTCAGGTTCAG
miR-21	CGGCGGTAGCTTATCAGACTGA
RNU6B	TTCCTCCGCAAGGATGACACGC

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
