# Peer review of "Sinomenine Inhibits Migration and Invasion of Human Lung Cancer Cell through Downregulating Expression of miR-21 and MMPs"

_ijms, 2020, doi:10.3390/ijms21093080_

Round 1

Reviewer 1 Report

I have reviewed this paper again and the authors have corrected/the three most important objections; therefore this manuscript is now recommended for publication

Reviewer 2 Report

The authors have addressed all my concerns.

This manuscript is a resubmission of an earlier submission. The following is a list of the peer review reports and author responses from that submission.

Round 1

Reviewer 1 Report

Good quality paper, sound methodology. Fig.2. Microscopic pictures of the migration/wound healing assay are poor and should be replaced by better pictures shedding more light on the scratch area. EMMPRIN should be referred to as EMMPRIN/CD147 as commonly known.

Sinomeneine effects in other systems/cells are much more pronounced; effects/IC50 values in the µM range are reported, for example, for U87 astrocytes and ACHN RCC cells - therefore, please discuss the low sensitivity of A549. It should be stressed that the results derived from single A549 may not be very representative for other NSCLC lines.

Reviewer 2 Report

The study is technically well performed. Major points that the authors need to address are as follows:

  1. Most of the experiments have been done in A549 cell line. Additional cell lines should be used to validate the key findings of the study.
  2. The molecular mechanism(s) by which Sinomenine can affect invasion/migration should be investigated in detail? For example, whether deletion of RECK gene by si-RNA can affect the observed anti-cancer effects of Sinomenine.
  3. Acute toxicity studies should be performed to establish the safety of Sinomenine.
  4. The authors should provide their own justification and relevance of the study. This will help the readers to understand the importance of the paper.
  5. A limited in vivo study will greatly enhance the impact of the findings.
  6. As transcription factor NF-κB plays an important role in regulating MMP expression, it will be interesting if the authors can observe the effect of Sinomenine on NF-κB signaling. Also, relevant literature such as (PMID: 27384261; PMID: 25083991; PMID: PMID: 18566231 etc must be included to improve the discussion part of the article.
  7. Typographical errors were found throughout the manuscript and should be corrected.